# The Role of HPV in Determining Treatment, Survival, and Prognosis of Head and Neck Squamous Cell Carcinoma

**DOI:** 10.3390/cancers14174321

**Published:** 2022-09-03

**Authors:** Imogen Sharkey Ochoa, Esther O’Regan, Mary Toner, Elaine Kay, Peter Faul, Connor O’Keane, Roisin O’Connor, Dorinda Mullen, Mataz Nur, Eamon O’Murchu, Jacqui Barry-O’Crowley, Niamh Kernan, Prerna Tewari, Helen Keegan, Sharon O’Toole, Robbie Woods, Susan Kennedy, Kenneth Feeley, Linda Sharp, Tarik Gheit, Massimo Tommasino, John J. O’Leary, Cara M. Martin

**Affiliations:** 1TCD CERVIVA Molecular Pathology Laboratory, The Coombe Women and Infants University Hospital, D08 XW7X Dublin, Ireland; 2Trinity St James Cancer Institute, Trinity College Dublin, D08 NHY1 Dublin, Ireland; 3Discipline of Histopathology, St. James’ University Hospital, Trinity College Dublin, D08 NHY1 Dublin, Ireland; 4Department of Pathology, Beaumont University Hospital, D09 V2N0 Dublin, Ireland; 5Department of Pathology, University Hospital Limerick, V94 F858 Limerick, Ireland; 6Department of Pathology, Mater University Hospital, D07 R2WY Dublin, Ireland; 7National Cancer Registry of Ireland, T12 CDF7 Cork, Ireland; 8Department of Pathology, St Vincent’s University Hospital, D04 T6F4 Dublin, Ireland; 9Department of Pathology, University Hospital Kerry, V92 NX94 Tralee, Ireland; 10Faculty of Medical Sciences, Newcastle University, Newcastle NE1 7RU, UK; 11Infections and Cancer Biology Laboratory, International Agency for Research on Cancer, 69008 Lyon, France; 12Dipartimento di Farmacia-Scienze del Farmaco, University of Bari, 70121 Bari, Italy

**Keywords:** HPV, head and neck cancer, HNSCC, oropharynx, oropharyngeal, human papillomavirus

## Abstract

**Simple Summary:**

Head and neck squamous cell carcinoma (HNSCC) incidence has been escalating in the last two decades, particularly in Western Europe and North America. Human papillomavirus (HPV) has been identified as the main culprit for this rise with significant implications for the treatment and outcomes of these patients. The purpose of this retrospective study was to investigate HPV’s impact on HNSCC patient outcomes in the Irish population, which has never been performed before. HPV positivity of HNSCC appears to be associated with improved survival in patients and highlights the importance of surgery, perhaps with less severe chemo-radiation therapy, in HPV-related HNSCC treatment.

**Abstract:**

Human papillomavirus (HPV) infection has been identified as a significant etiological agent in the development of head and neck squamous cell carcinoma (HNSCC). HPV’s involvement has alluded to better survival and prognosis in patients and suggests that different treatment strategies may be appropriate for them. Only some data on the epidemiology of HPV infection in the oropharyngeal, oral cavity, and laryngeal SCC exists in Europe. Thus, this study was carried out to investigate HPV’s impact on HNSCC patient outcomes in the Irish population, one of the largest studies of its kind using consistent HPV testing techniques. A total of 861 primary oropharyngeal, oral cavity, and laryngeal SCC (OPSCC, OSCC, LSCC) cases diagnosed between 1994 and 2013, identified through the National Cancer Registry of Ireland (NCRI), were obtained from hospitals across Ireland and tested for HPV DNA using Multiplex PCR Luminex technology based in and sanctioned by the International Agency for Research on Cancer (IARC). Both overall and cancer-specific survival were significantly improved amongst all HPV-positive patients together, though HPV status was only a significant predictor of survival in the oropharynx. Amongst HPV-positive patients in the oropharynx, surgery alone was associated with prolonged survival, alluding to the potential for de-escalation of treatment in HPV-related OPSCC in particular. Cumulatively, these findings highlight the need for continued investigation into treatment pathways for HPV-related OPSCC, the relevance of introducing boys into national HPV vaccination programs, and the relevance of the nona-valent Gardasil-9 vaccine to HNSCC prevention.

## 1. Introduction

Prognosis and survival for head and neck squamous cell carcinoma (HNSCC) are generally poor. Approximately half of all patients with HNSCC have advanced-stage disease at the time of diagnosis, with an expected 5-year survival rate between 10% to 40% [1]. This is mostly attributed to the fact that diagnosis of HNSCC is frequently delayed because symptoms for which patients will seek medical attention such as pain, dysphagia, and shortness of breath occur late in the stage of the disease [2].

HPV-positive HNSCC, and more specifically oropharyngeal squamous cell carcinoma (OPSCC), has a unique relationship to diagnosis, prognosis, and treatment. These tumors generally present with a more advanced clinical stage, with a higher nodal category [3,4], despite lower tumor extent [4,5], and have different tendencies for extracapsular spread and perineural invasion [6]. Tonsil squamous cell carcinoma (SCC) in general is known to present with early lymph node metastases [7] and it may simply be that the anatomy of the site itself facilitates the early spread and depth of invasion [8].

Despite more advanced presentation, improved survival, which is consistently higher than 30% [9], is evident in HPV-related OPSCC [8,10,11,12,13,14], irrespective of treatment modality [5,15,16,17,18,19,20,21]. The improved prognosis and response to treatment holds true for all indicators of HPV-positivity including seropositivity, mRNA, oncoprotein expression, and viral load and copy number [22]. It also remains salient in the case of HPV-positive OPSCC biomarkers, including p16, p53, EGFR, and Bcl-xL [22,23].

For most patients with high-risk, resected HNSCC, the standard treatment constitutes adjuvant radiation therapy with high doses of cisplatin. This course of treatment appears to work well for HPV-positive tumors. Adjuvant chemoradiation therapy with one dose of weekly cisplatin had 3-year overall survival rates of 86% and 91% and 3-year recurrence free survival of 82% and 84% in one study, suggesting that cisplatin is a good treatment for HPV-positive OPSCC to preserve survival and minimize toxicity [24,25].

Given this positive response to therapies [5,26,27,28,29], de-escalation of therapy might be appropriate for these HPV-positive HNSCCs. This is particularly important given the long-term consequences and associated morbidities amongst those patients who do survive. Though patients express gratitude for the success of their treatments, many suffer from difficulty swallowing, breathing, and speaking, chronic pain, osteoradionecrosis, hypertension, pneumonia, dysphagia, weight loss, malnutrition, dental issues, and third-degree burns. These are acute hindrances to the quality of the rest of their lives.

Despite extensive reports in the literature, most studies regarding the differential prognosis and treatment modalities of HPV-related and HPV-unrelated HNSCC analyze fewer than 300 cases [13,14,30]. When pooled for meta-analysis [21], the definition of what constitutes an HPV-positive case is heavily dependent upon the HPV indicator chosen and the technology used, which varies by study. Furthermore, there is an evident gap in the literature regarding the Irish population’s HPV-related and HPV-unrelated HNSCC survival. This gap provides a unique opportunity to study historical samples from a period before the widespread assessment of HPV status determined patient management, allowing for a prognostic comparison without management as a confounder.

In the context of the worldwide meta-analysis being conducted by the International Agency for Research on Cancer (IARC) on the subject (HPV-AHEAD, http://hpv-ahead.iarc.fr, accessed on 1 July 2022), we collected over 1115 FFPE HNSCC samples from six different hospital sites across Ireland, aiming to determine the relationships between HR HPV status, treatment scheme, and survival. We also standardized the definition of an HPV-positive case, using DNA alone detected by an extremely sensitive Multiplex PCR Luminex technology. In this article, we describe the first results of comprehensive survival analysis of HPV DNA-positive and negative OPSCC, oral cavity SCC (OSCC), and laryngeal SCC (LSCC) diagnosed in Ireland between 1994 and 2013. These patient samples were not routinely analyzed for HPV in this time period.

## 2. Materials and Methods

### 2.1. Specimen Collection

Through the National Cancer Registry of Ireland (NCRI) database, an initial incident population of 5767 OPSCC, OSCC, and LSCC was identified. ICD10 codes used to define these sites were in line with the most up-to-date World Health Organization (WHO) classifications [31]. Strict inclusion criteria were applied to this initial database including that cases should be: (i) archival primary tumors (ii) diagnosed between 1994 and 2013 (iii) purely and histologically confirmed squamous cell carcinoma (iv) plentiful enough to conduct all necessary analyses. A total of 1115 specimens fulfilling these criteria based on review of associated pathology reports were retrieved. All cases were re-cut, H+E stained, and re-reviewed by pathologists to confirm there was still tumor remaining in specimens selected for HPV analysis. Following this, 861 cases remained for molecular testing. Ethical approval for the study was obtained from 11 different research ethics committees across Ireland representing 14 different hospitals, 6 of which were ultimately included and sourced for specimens. The study’s use of only archival material ensured no contact with patients, and cases were anonymized by the NCRI using a random study number. Consent was obtained from patients still alive through the NCRI, and a waiver on consent was obtained from the different RECs at that time for use of anonymized material from deceased patients. The blocks were used in the manner detailed below to generate the coming results.

### 2.2. Preparation of the Tissue Sections

FFPE tissue sections were cut in the Trinity College Dublin CERVIVA Molecular Pathology Research Laboratory based at the Coombe Women and Infants’ University Hospital (CWIUH), Dublin, Ireland in accordance with the HPV-AHEAD sectioning protocol [32] and using a Leica^®^ RM2135 (Leica Biosystems, Wetzlar, Germany) [33] instrument. Briefly, for DNA analysis, five sections were cut in order from S1 to S5. S1 and S5 were cut for H+E slides (Leica^®^ Bond Plus charged slides (Leica Biosystems, Wetzlar, Germany) [34]) at 5 μm to confirm the presence of appropriate tumor tissue for all sections used for molecular testing. S2, S3, and S4 were cut for HPV DNA analysis at 10 μm and placed in a 1.5 mL DNase/RNase-free 1.5 mL Micro tube (Sarstedt, Wexford, Ireland) [35]. To minimize the risk of cross-contamination during sectioning, the microtome was cleaned extensively between each FFPE block using ethanol 70% and DNA ZAP^TM^ (ThermoFisher^TM^ Scientific, Waltham, MA, USA) [36]. A new blade was used for each block and after 10 tumor tissue blocks were cut, sections were generated from an empty paraffin block and a known HPV DNA-positive block comprised of SiHa or HeLa cells. These were all included in the DNA analyses.

### 2.3. Histological Analysis

Two H+E slides were generated for every block, resulting in 2230 slides from the original 1115 FFPE blocks retrieved. These were all analyzed by a Pathology Review Board (PRB) comprised of 6 pathologists based in St. James’ University Hospital, Dublin, Ireland, and the CWIUH, Dublin, Ireland. A subset of 20% of cases was reviewed a second time by another member of the Pathology Review Board to confirm diagnosis, with little to no divergence in assessment. Only cases with relevant tumor tissue in both associated H+Es were brought forward for molecular testing, resulting in 861 cases ultimately used in the study.

### 2.4. HPV DNA Genotyping

At the Trinity College Dublin CERVIVA Molecular Pathology Laboratory at the CWIUH, HPV DNA was extracted from pooled sections S2 to S4 using a 250 μL of digestion buffer (10 mM Tris/HCl pH 7.4, proteinase K 0.5mg/mL, and Tween 20 0.4%) and a 2 h incubation at 56 °C. Water samples were also included to signal contamination at any stage of DNA preparation.

HPV DNA was detected by a type-specific multiplex genotyping assay developed previously [37] and based in IARC, Lyon, France. This method combined Multiplex PCR and bead-based Luminex Technology (Luminex Corp, Austin, TX, USA) [38,39]. The Multiplex PCR uses HPV type-specific primers targeting the E7 region of 21 genotypes. A total of 19 of these are high-risk (HR) or possible HR (pHR) and include HPV16, 18, 26, 31, 33, 35, 39, 45, 51, 52, 53, 56, 58, 59, 66, 68a, 68b, 70, 73, and 82. The remaining 2 are low-risk types HPV6 and 11. Detection limits of the PCR range from 10 to 1000 copies per reaction. To control for quality of the template DNA, two primers for amplification of the beta-globin were included. The slight modification of the protocol described previously [32,40] for the amplification of shorter (~100 bp) fragments for ten HPV genotypes (16, 18, 31, 33, 35, 52, 56, 66, 6, 11) and 117 bp for β-globin were applied.

Post PCR, 10 μL of each reaction mixture was analyzed by Multiplex HPV genotyping using Luminex xMAP^®^ technology (Luminex Corp, Austin, TX, USA) [38,39] as described previously [41,42]. The aforementioned empty paraffin blocks, known positive blocks, and water samples were analyzed alongside all specimens blindly, with no evidence of contamination at any stage.

### 2.5. Statistical Analysis

Independent variables, survival, and treatment data were provided for each patient case by the NCRI including all those in Table 1, Table 2 and Table 3. The NCRI provided anonymized study numbers to the researcher that linked all HPV analyses the researcher performed to the associated characteristics in the national database. These variables were then used to compare and contrast survival, prognosis, and treatment administered between HPV-related and HPV-unrelated groups.

Statistics were generated using IBM SPSS Statistics Version 25 (IBM, Chicago, IL, USA), XLSTAT 2019.1.3, and Microsoft Excel Version 16.25. Overall and cancer-specific survival for all oropharyngeal, oral cavity, and laryngeal cancer, and within each sub-site, based on HR HPV status was assessed by Kaplan–Meier curves and log-rank test. Additional Cox proportional hazard statistics were generated to confirm Kaplan–Meier results. The relationship between treatment and HPV status for all cases and for each sub-site was evaluated using chi-square statistics and Fisher’s exact tests in cases where expected counts fell below 5. The cohort of patients assessed for treatment was limited to those who received treatment of any kind within 12 months of diagnosis as these were patients of interest. Predictors of overall and cancer-specific survival were evaluated individually by univariable Cox proportional hazard models. For variables from which more than 10% of data were missing, “missing” was included as a category of its own as is convention in the literature to account for or detect any bias responsible for significance. For those variables with between 0% to 10% missing data, cases with missing data were excluded for univariable and multivariable analyses. Those variables significant in univariable analysis were brought forward for multivariable analysis. All significant variables by univariable models were included in the initial multivariable model. The least significant predictor was then taken out, and the model was run again. The least significant predictor was again taken out, and the model was run again. This continued until all variables remaining in the model proved significantly predictive of survival and risk of death, or until taking another variable out rendered the model as a whole insignificant.

## 3. Results

### 3.1. Summary of the Study Population

To contextualize the study population, its basic demographic characteristics are presented in Table 4 below.

A basic summary of the number of HPV positive and negative cases by sub-site is provided below in Table 5.

### 3.2. Survival by HPV DNA Status

Figure 1 and Figure 2 show the result of the Kaplan–Meier analysis of overall and cancer-specific survival for the population stratified by HPV status, respectively. There was significantly worse survival for the HPV-negative group than for the HPV-positive group (Log-rank: Chi-square = 12.593, 1 d.f., *p* < 0.0001). Cox proportional hazard model for HPV status and overall survival confirmed the increased risk of death for HPV-negative patients (HR = 0.372, 1 d.f., *p* < 0.0001). Figure 2 mirrors findings in Figure 1, showing better survival for HPV-positive cases than HPV-negative cases (Log-rank: Chi-square = 4.582, d.f. = 1, *p* = 0.032). Cox proportional hazard model seconded the significantly increased risk of cancer-specific death in the HPV-negative group (HR = 0.257, SE = 0.122, *p* = 0.035).

Figure 3 and Figure 4 showcase the Kaplan–Meier analysis of overall and cancer-specific survival stratified by HPV status for the oropharyngeal sub-site alone, respectively. Much like for all cases, there was significantly worse prognosis for HPV-negative cases (Overall-Log-rank: Chi-square = 17.017, 1 d.f., *p* < 0.0001; Cancer-specific-Log-rank: Chi-square 11.902, 1 d.f., *p* = 0.001). Cox proportional hazard model confirmed this finding (HR = 0.659, SE = 0.165, *p* < 0.0001; HR = 00.620, SE = 0.185, *p* = 0.001).

No significant differences in survival by HPV status were noted for LSCC or OSCC sub-sites alone.

### 3.3. The Relationship between Treatment Modality and HPV Status

There was significant relationship between HPV status and treatment modality administered (Chi-square = 49.732, 4 d.f., *p* < 0.0001). HPV-positive patients were almost twice as likely to be treated more aggressively with all three treatment modalities (surgery/radiotherapy/chemotherapy) than HPV-negative cases. More HPV-negative patients were treated with surgery or radiotherapy alone and almost three times as many HPV-positive patients were treated chemically with radiotherapy/chemotherapy than HPV-negative patients. These patterns were mimicked within the population of oropharyngeal patients alone (chi-square = 14.401, 4 d.f., *p* = 0.006). No associations were seen within oral cavity cases alone (chi-square = 7.837, 4 d.f., *p* = 0.098) but were within the laryngeal case population alone (Fisher’s exact = 12.423, *p* = 0.007). These differences emanated from the disproportionate treatment of HPV-negative cases with radiotherapy alone, and the more frequent treatment of HPV-positive patients with all three modalities.

### 3.4. The Relationship between HPV Status, Treatment Modality, and Survival

Given the significant relationship between HPV status and treatment modality, and HPV status and survival only for OPSCC, the relationship between all three variables for OPSCC alone is presented. For HPV-positive cases in the oropharynx, there was a significant difference in overall survival by treatment type (Log-rank = 10.481, 4 d.f., *p* = 0.033) (n = 80) (Figure 5). Cox proportional hazard model reflected this significance for all treatments (HR = 1.166, 0.579, −0.055, 0.096, SE = 0.460, 0.443, 0.803, 0.559, *p* = 0.049, 0.011, 0.192, 0.946, 0.864). Surgery alone and all three treatment modalities maximized overall survival for HPV-positive OPSCC patients. Treatment with radiotherapy alone was associated with decreased survival. For cancer-specific survival amongst HPV-positive oropharyngeal cases, there was a significant difference by treatment type (Log-rank = 11.398, 4 d.f., *p* = 0.022). Figure 6 showcases this difference. Indeed, this significance was reflected by Cox proportional hazard model, with surgery having the lowest risk of death, followed by surgery/radiotherapy and surgery/radiotherapy/chemotherapy, followed by radiotherapy/chemotherapy, with radiotherapy having the worst risk of death (HR = 1.186, 0.490, −0.567, −0.137, SE = 0.496, 0.484, 1.082, 0.649, *p* = 0.039, 0.017, 0.312, 0.600, 0.833).

For HPV-negative oropharyngeal cases, there was no difference in overall or cancer-specific survival by treatment type, something also reflected by the Cox proportional hazard model.

### 3.5. Modeling Predictors of Survival

Table 6, Table 7, Table 8, Table 9, Table 10 and Table 11 summarize the variables significantly predicting overall and cancer-specific survival, respectively, for OPSCC, OSCC, and LSCC populations.

## 4. Discussion

The key finding of the present analysis was that both overall and cancer-specific survival were significantly improved for HPV-positive cases in all oropharyngeal, oral cavity, and laryngeal SCC grouped together. This relationship emanated, however, from the OPSCC population alone. This is reflective of most studies in the literature [14,30,43]. This is likely given that HPV-related patients, being younger, are less likely to have had significant exposure to tobacco, marijuana, alcohol, diabetes, chronic obstructive pulmonary disease, anxiety disorders, and major depression. The most at-risk populations are thus those with the best immune ability to combat HPV-related disease. Furthermore, the current results support the notion that the viral origins of HPV-positive tumors, accompanied by their expression of viral oncoproteins and related HPV-positive tumor antigens at sites of huge immune and lymphatic activity likely attracts a more aggressive and specific immune response that improves both overall and cancer-specific survival [44,45]. Younger patients are also more likely to better survive harsh treatments and their potential side effects, which is particularly important in the case of HPV-positive tumors in this population given that they were more likely to be treated harshly than HPV-negative patients.

Figure 5 and Figure 6 show very clearly that survival amongst HPV-positive OPSCC patients was longest amongst those treated with surgery alone. Figure 6 shows over 70% cancer-specific survival rates after 10 years for patients treated with surgery alone, with those treated by all three modalities and surgery/radiotherapy following closely behind.

These results highlight the importance of surgical intervention for HPV-positive OPSCC, with treatment approaches not involving surgery seeing very poor survival. The findings are extremely promising in terms of the potential for de-escalation of treatment for these patients. This is indicative of the chance that HPV-related OPSCC presents for drastically improving the quality of life for patients by avoiding the administration of extremely harsh treatments and the long-term side effects that accompany them including difficulties swallowing, breathing, and speaking, chronic pain, osteoradionecrosis, hypertension, pneumonia, dysphagia, weight loss, malnutrition, dental issues, and third-degree burns [46,47] along with the increased risk of heart disease and failure [48,49], risk of another (non-recurrence) primary tumor at another site [50,51,52,53], and complications due to immunosuppression.

The present data thus supports the notion that robotic trans-oral resection (TOR) alone yields extremely good results for HPV-related patients [54,55,56] regardless of stage and posits that this kind of non-chemical curative approach gives patients better functional outcomes [57,58,59] may be the way forward. Other studies are in agreement that TOR without adjuvant therapy is often an adequate treatment for HPV-related OPSCC, with anywhere between 48% to 74% of patients not requiring chemotherapy after TORs [54,55,60]. This said, it is understandable that patients may feel more comfortable being treated with more than just surgery, with studies showing that nearly 70% of patients are not willing to risk a 5% or less drop in survival likelihood to switch from chemoradiation to radiation alone after surgery [61]. In the present population, this 5% drop in survival is not evident amongst HPV-positive OPSCC patients, with surgery alone seeing better survival than surgery/radiotherapy, and the margin between surgery/radiotherapy and all three treatments being minimal (Figure 6). This is something that may give patients more incentive to opt for less harsh schemes. Nonetheless, many trials currently underway are based on the suggestion that surgery with de-escalated radiotherapy yields maximal survival with decreased morbidity and associated side effects [62,63,64], a scheme that might satisfy survival outcomes, minimize side effects, and ensure patient peace of mind simultaneously.

Despite positive indications of de-escalation potential in HPV-positive OPSCC, the present analysis indicates that HPV-positive HNSCC and OPSCC were more likely to be treated harshly than their HPV-negative counterparts. The population that might have benefited most from less severe treatment schemes was thus the population being treated most severely. The present data and the literature explain that this irony is due to the later stage at which HPV-related OPSCC is diagnosed [9,43,65,66]. Specifically, they disproportionately present at Stage IV due to the late N stage according to the 5th edition AJCC guidelines relevant to this population between 1994 and 2013 [67]. The current analysis posits therefore that the new 2017 8th edition AJCC guidelines [68] updated for the oropharyngeal sub-site alone, reflecting the role of HR HPV, are very highly relevant. This is especially true since neither the N stage nor the TNM stage were significant predictors of overall survival in OPSCC. TNM stage was barely significant in predicting survival in OPSCC at the cancer-specific level (Table 6), and the N stage remained insignificant. This implies that the nodal and cumulative staging of the older staging systems were not accurate assessors of the aggressivity of these tumors, likely due to the unique features of HPV-related tumors in this region. Those HPV-related cases diagnosed as Stage IV before 2017 will now be downgraded to at least Stage III if not even Stage I due to adjustments in the N stage relating to nodal metastasis. It is very likely that the consequent down-grading of the stage in OPSCC will act as a de-escalation mechanism of its own, implicating less severe treatment requirements from the moment the cancer is diagnosed. On the basis of pending results of current trials [64], the clinical context may need to adapt.

To note is that it is increasingly recognized that HPV-negative oropharyngeal cancer represents a treatment-resistant entity and a distinct therapeutic challenge. Comparison among treatment modalities for oropharyngeal cancer has been limited, and very few studies have evaluated differences based on the HPV subtype. Our findings that HPV-negative cases were treated less aggressively than HPV-positive tumors may in fact just be reflective of clinical practices at that time, or indeed may be related to the patient demographic where smoking and alcohol use may play a role in treatment responses. Further prospective studies into outcomes for HPV-negative tumors are warranted.

Two caveats to this study’s support of de-escalation should be noted. First, there were smaller sample sizes available when sub-dividing all 861 cases into their sub-site, HPV status, and treatment groups. Targeted sampling of OPSCC alone is needed for further confirmation of these promising findings. Second, the analysis also emphasizes that in terms of potential de-escalation, it would be unethical to make treatment decisions for these patients, or their negative counterparts, based solely on HPV status. For oropharyngeal, oral cavity, and laryngeal SCC, multivariable predictors of the overall risk of death did include HPV negativity, but HPV status was not confounded by other patient characteristics including older age and current smoker status (Table 6, Table 7, Table 8, Table 9, Table 10 and Table 11). For OPSCC, HPV positivity was predictive of decreased risk of death at the overall and cancer-specific levels (Table 6 and Table 7). However, HPV was not confounded by age or social deprivation. It also did not predict the risk of death for any survival in LSCC and OPSCC (Table 8, Table 9, Table 10 and Table 11).

The findings indicate that the oropharynx is the sub-site in which HPV-related tumors occur and that it is therefore the region for which any HPV-related treatment alterations should be made. They also highlight that though HPV-related tumors are already significantly associated with younger aged patients [43,64,65] and never-/ex-smokers [69,70], it would be extremely prudent to select patients who might benefit from de-escalation based on not only HPV-positive status but also on other survival-maximizing characteristics at both the cancer-specific and overall levels. Table 12 summarizes those patient characteristics that the present multivariable analysis indicates as being stereotypically HPV-driven cases and might benefit from de-escalated treatment.

This collection of patient characteristics has recently been recognized in the literature as the only group of oropharyngeal, oral cavity, and laryngeal SCC patients for which de-escalation of treatment is acceptable. In fact, several of the ongoing trials regarding de-escalation only include patients meeting these criteria to assure no jeopardizing of patient safety [62,63,64,71], but also to target the group that will likely benefit most from less severe treatment. This said, very recent publications from the largest de-escalation trials for these patients prove a cautionary tale. DE-ESCALATE and RTOG 1016 trials have thus far shown that omitting cisplatin or substituting it shows a detrimental impact on survival in HPV-positive OPSCC [72]. This said, several non-randomized phase II cohort studies attempting lower radiotherapy doses have shown promising results despite lacking control arms [62,73,74]. The details of the definition of “de-escalation” are thus still being clinically determined and much like these newly published studies suggest, no amendments to current treatment regimes for HPV-positive patients should be implemented in the clinic until this time [72].

With respect to Table 6 and Table 7, it should also be noted that there is still a need to distinguish clinically significant HR HPV infections from transient ones. While in this analysis HR HPV DNA was used to determine HPV-related status, many trials only use p16 as a representative biomarker of an active HPV infection [64]. Neither of these alone is entirely satisfactory in the clinical context given the potential for transient HR HPV infections, and the expression of p16 regardless of HPV status. In fact, HPV DNA may be misleading even if other patient characteristics are suggestive of a classically HPV-related case. In the clinic, these kinds of risks resulting in the potential under-treatment of patients cannot be taken. Further specification of “HPV positivity” as a necessary characteristic for de-escalation will likely make treatment decisions and thus survival determinations even more accurate. Pairing p16 with HR HPV DNA [32,75], or simply using HPV mRNA [76], represents mechanisms to refine this process in the clinic, though the present HR HPV DNA data are a resounding start.

Thus, this study population is further proof of the suggestion that HPV-positive oropharyngeal, oral cavity, and laryngeal SCC, and more specifically, OPSCC alone, is a better-surviving cancer than its HPV-negative counterparts. Furthermore, though more than simply surgery is and will continue to be necessary to treat some late-stage patients, the benefit of treating all HNSCC patients indiscriminately with all three modalities is questionable. The data also indicate that other indicators of clinically relevant HPV infections including patient characteristics will play a significant role in determining treatment options for oropharyngeal, oral cavity, and laryngeal SCC in addition to, and not instead of sensitive HR HPV detection.

Importantly, the diverse treatment, survival, and HPV characteristics observed in the present data converge and point to the crucial nature of prevention and early detection in oropharyngeal, oral cavity, and laryngeal SCC if survival, overall and cancer-specific, is to be maximized. All HNSCC are overwhelmingly behaviorally driven cancers, whether by exposure to HPV and/or to smoking (and likely alcohol).

For OPSCC specifically, data are still emerging on the impact of the quadra- and nona-valent Gardasil vaccines on HR HPV prevalence in the oral cavity [77], though preliminary data from cervical trials testing oral rinses show that HPV16/HPV18 prevalence is lower in vaccinated groups compared to control groups, with an estimated efficacy of 93.3% for HPV16/18 [78]. Predictive modeling studies also suggest that with a 50% vaccination uptake and 50% vaccine efficacy, the vaccination of young boys for the prevention of HPV-related OPSCC would be cost-effective [77,79]. The need for more data is evident, but the systemic nature of vaccines logically suggests that the administration of the vaccine in early adolescence should be as effective in preventing HNSCC as it is in the cervical context. The FDA has recently approved the indication of Gardasil-9 to include the prevention of oropharyngeal and other head and neck cancers caused by HPV types 16, 18, 31, 33, 45, 52, and 58, and while in Europe, Gardasil-9 is yet licensed this indicates that many governments including Ireland have now expanded the public HPV vaccination scheme to include boys is, therefore, encouraging as an HNSCC prevention strategy.

Lastly, the present analysis underlines the urgent need for effective and systematic HNC screening tools. Early detection of those SCC that do go on to develop despite preventative measures is tantamount to prolonging overall survival, no matter how promising or poor cancer-specific survival is and regardless of HPV status. For HPV-unrelated HNSCC in this analysis, especially in the larynx, diagnosis at a later TNM stage was the only predictor of cancer-specific survival after adjustment for other variables (Table 11). Efforts are currently being made to investigate the best ways to sample tissue from the oral site, but it is made difficult by the region’s confined nature and the dense, complex network of MALT tissues that line it [80,81]. Mobile microscopy with a simple brush biopsy has shown to be an effective screening mechanism for oral cavity cancer, even in low-resource areas [82], but such a sampling method is not ideal for the deep, hidden, crypts of the oropharynx. The role that HPV might play in this screening is also uncertain, though monitoring systems such as those established in the cervical case [83,84] are a promising way of catching HR HPV patients who, perhaps even after vaccination, go on to develop lesions.

## 5. Conclusions

In all, pairing early detection with preventative mechanisms and curative approaches suitable to the tumor and patient characteristics will render oropharyngeal, oral cavity, and laryngeal SCC an imminently manageable and rare disease. These public health and clinical measures will ultimately mean huge cost savings, and more importantly, the difference between life and death for potential and current oropharyngeal, oral cavity, and laryngeal SCC patients.

## Figures and Tables

**Figure 1 cancers-14-04321-f001:**
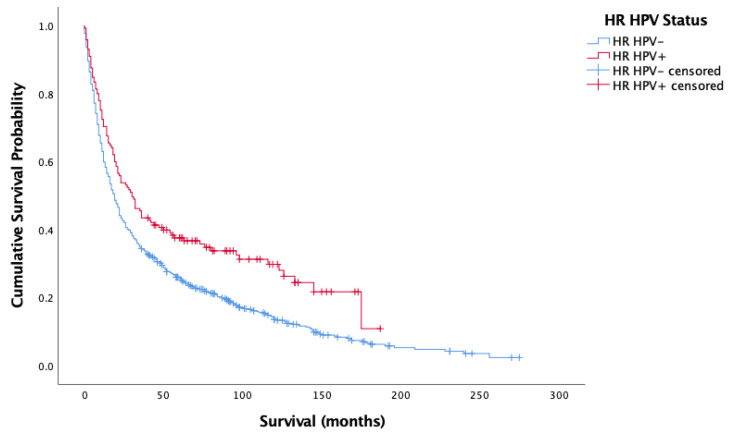
Kaplan–Meier analysis of overall survival in months based on HR HPV status for oropharyngeal, oral cavity, and laryngeal cancer (*n* = 861).

**Figure 2 cancers-14-04321-f002:**
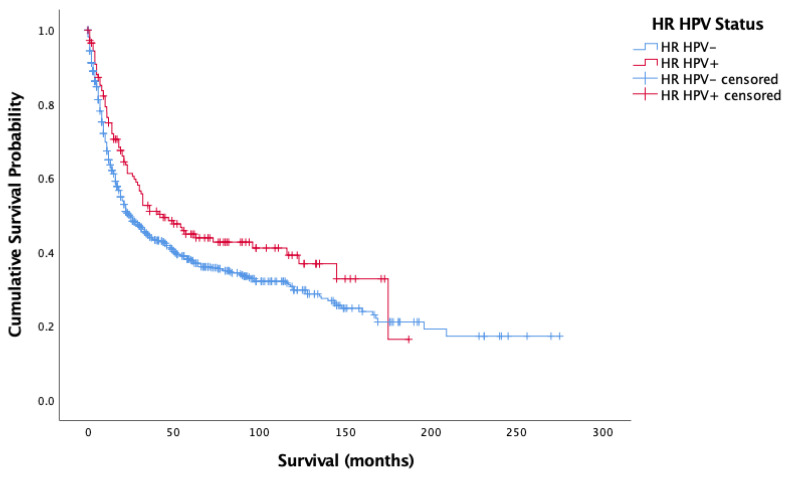
Kaplan–Meier analysis of disease-specific (cancer-specific) survival in months based on HR HPV status for oropharyngeal, oral cavity, and laryngeal cancer (*n* = 861).

**Figure 3 cancers-14-04321-f003:**
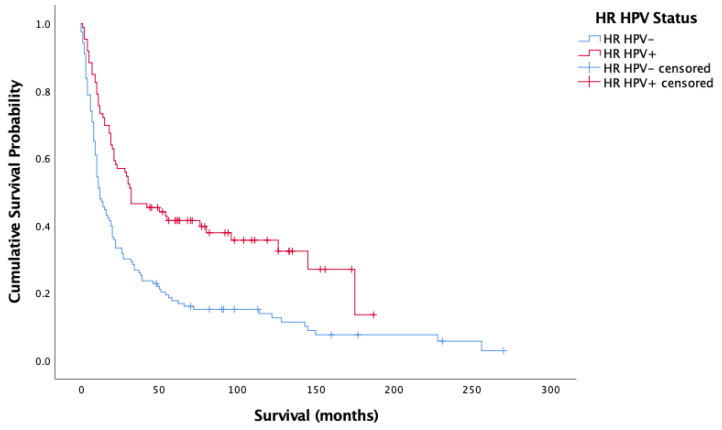
Kaplan–Meier analysis of overall survival in months based on HR HPV status for oropharyngeal cancer (*n* = 209).

**Figure 4 cancers-14-04321-f004:**
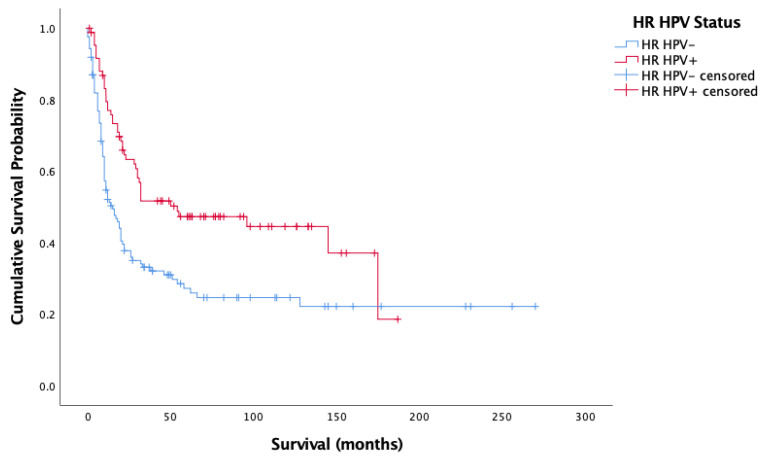
Kaplan–Meier analysis of disease-specific (cancer-specific) survival in months based on HR HPV status for oropharyngeal cancer (*n* = 209).

**Figure 5 cancers-14-04321-f005:**
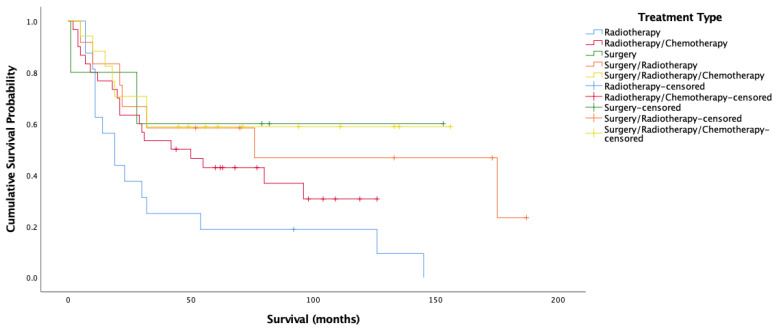
Kaplan–Meier analysis for overall survival amongst HPV-positive oropharyngeal cancer stratified by treatment type (*n* = 80).

**Figure 6 cancers-14-04321-f006:**
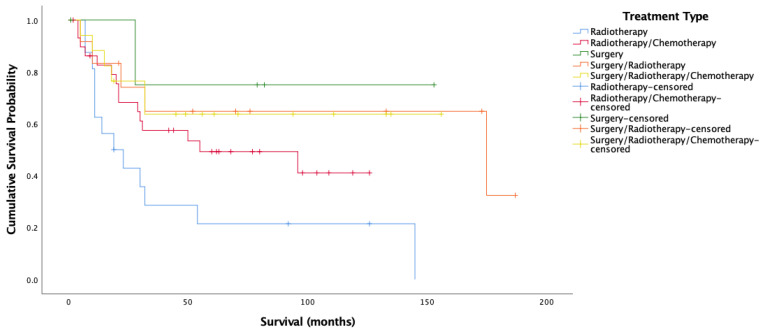
Kaplan–Meier analysis for cancer-specific survival amongst HPV-positive oropharyngeal cancer stratified by treatment type (*n* = 80).

**Table 1 cancers-14-04321-t001:** Independent variables made available by the NCRI for the population of the study and notes on any adjustments made for the purposes of the analysis.

Variable Code	Meaning	Variable Definition	Notes on Adjustments
SEX	Sex of patient	Sex of patient	N/A
AGE	Integer age at date of diagnosis	Integer age at date of diagnosis	Age was assessed both continuously and based on age younger than or equal to, and older than age 50. Only continuous age was brought forward for multivariable analysis where relevant.
SMOKER_ ID	Smoking status	Indication of current, ex-, or never-smoked behavior	N/A
GRADE	Grade of primary tumor	Poorly, moderately, well-, or un-differentiated grade of tumor	Only 2 undifferentiated cases were detected in the population. These were excluded after distribution was determined for all grade statistics generated to avoid skew in results.
T5	T stage	T category of stage (5th edition for cases diagnosed up to 2013) derived from best available clinical or pathological T data	Due to low frequencies for sub-stages, these were combined to yield the following T stage categories: T1, T2, T3, T4.
N5	N Stage	N category of stage (5th edition for cases diagnosed up to 2013) derived from best available clinical or pathological N data	Due to low frequencies for sub-stages, these were combined to yield the following N stage categories: N0, N1, N2, N3. N2 and N3 were also combined due to extremely low numbers of N3 patients.
M5	M Stage	M category of stage (5th edition for cases diagnosed up to 2013) derived from best available clinical or pathological M data	N/A
TNM5	TNM Stage	TNM stage (5th edition for cases diagnosed up to 2013) derived from best available clinical or pathological data	Due to low frequencies for sub-stages of Stage IV, TNM stages were combined to yield the following categories: Stage I, II, III, IV.
COUNTY_ RES	County of residence	County of residence of patient at time of diagnosis	Due to low frequencies for many counties, county was assessed based on both residence in counties with large urban centers (Dublin/Limerick/Cork) and residence in or outside Dublin.
DEPRIV_ POBAL_ 2011	Socio-economic status/Social Deprivation Score	Pobal index of deprivation from 1 to 5 for 2011 patient’s Electoral Division (ED) of residence at diagnosis re-expressed as quintiles of 2011 population	Social deprivation score was categorical on a scale of 1 to 5, with 5 being the most deprived. It was assessed both categorically and as a continuous variable.
MARITAL	Marital status	Indication of single, separated, widowed, or divorced status of patient	Divorced and separated individuals were grouped together due to similarity in status and low numbers of divorced patients.

**Table 2 cancers-14-04321-t002:** Variables regarding patient treatment provided by the NCRI. These variables were used individually and in combination with one another for the analysis.

Variable Code	Meaning	Definition
Chemo_1y	Chemotherapy	Binary indication of whether or not patient was treated with chemotherapy targeting the cancer within 1 year of diagnosis.
Radio_1y	Radiotherapy	Binary indication of whether or not patient was treated with radiotherapy targeting the cancer within 1 year of diagnosis.
Surg_1y	Surgery	Binary indication of whether or not patient was treated with surgery targeting the cancer within 1 year of diagnosis

**Table 3 cancers-14-04321-t003:** Variables regarding patient survival provided by the NCRI.

Variable Code	Meaning	Variable Definition
VITAL_STAT	Overall survival	All-cause vital status of patient (0 alive or 1 dead) at common censoring date based mainly on death-certificate matching.
VITAL_CAN	Disease-specific (cancer-specific) survival	Cause-specific vital status (0 alive or died of other cause or different cancer or 1 died from the cancer of interest) at common censoring date.
SURVIVAL_MONTHS	Survival in months	Number of complete months from diagnosis of a specific tumor to common censoring date.

**Table 4 cancers-14-04321-t004:** Summary of the study population.

Variable/Characteristic	Sub-Set of Variable	Proportion/Mean/Median
Sex	Male	(661/861) = 76.8%
	Female	(200/861) = 23.2%
Age (Continuous)		Mean = 63.30 (CI: 62.52, 64.08) Median = 63.00
Age ≤ 50	≤50	(121/861) = 14.1%
	>50	(740/861) = 86.9%
Smoking Status	Current smoker	(479/861) = 55.6%
	Ex-smoker	(110/861) = 12.8%
	Never smoked	(156/861) = 18.1%
	Unknown	(116/861) = 13.5%
Sub-site	Oropharynx	(209/861) = 24.3%
	Oral Cavity	(331/861) = 38.4%
	Larynx	(321/861) = 37.3%
Grade	Well-differentiated	(88/861) = 10.2%
	Moderately differentiated	(475/861) = 55.2%
	Poorly differentiated	(187/861) = 21.7%
	Un-differentiated	(2/861) = 0.2%
	Unknown	(109/861) = 12.7%
TNM Stage	Stage I	(119/861) = 13.8%
	Stage II	(126/861) = 14.6%
	Stage III	(133/861) = 15.4%
	Stage IV	(376/861) = 43.8%
	Unknown	(107/861) = 12.4%

**Table 5 cancers-14-04321-t005:** HPV DNA prevalence for oropharyngeal, oral cavity, and laryngeal cancer diagnosed in Ireland between 1994 and 2013.

Sub-Site	Fraction	Prevalence
Oropharynx	86/209	41.1%
Oral Cavity	36/331	10.9%
Larynx	25/321	7.8%
All	147/861	17.1%

**Table 6 cancers-14-04321-t006:** Patient and tumor characteristics significantly predicting overall survival amongst oropharyngeal cancer patients by multivariable Cox proportional hazard model (*n* = 189).

Variable/Factor	Statistic	Increased Risk of Death
Age (Continuous)	HR = 0.020 SE = 0.009 *p* = 0.029	Older age
HR HPV Status	HR = 0.737 SE = 0.190 *p* < 0.0001	HPV negativity
T Stage	Base comparison to: T4 (vs. T3, T2, T1, Missing) HR = −0.709, −1.548, −0.604, −0.593 SE = 0.273, 0.273, 0.221, 0.247 *p* < 0.0001, 0.009, 0.0001, 0.006, 0.016	T4 > T3, T2, Missing > T1
M Stage	Base comparison to: M1 (vs. M0, Missing) HR = −1.049, −1.198 SE-0.316, 0.313 *p* = 0.001, 0.001, 0.0001	M1 > M0, Missing

**Table 7 cancers-14-04321-t007:** Patient and tumor characteristics significantly predicting disease-specific (cancer-specific) survival amongst oropharyngeal cancer patients by multivariable Cox proportional hazard model (*n* = 209).

Variable/Factor	Statistic	Increased Risk of Death
Age (Continuous)	HR = 0.039 SE = 0.012 *p* = 0.002	Older age
HR HPV Status	HR = 0.937 SE = 0.247 *p* < 0.0001	HPV negativity
Deprivation Score	HR = 0.165 SE = 0.064 *p* = 0.010	More deprived
TNM Stage	Base comparison to: T4 (vs. T3, T2, T1, Missing) HR = −0.694, −1.615, −0.564, −0.540 SE = 0.327, 0.598, 0.305, 0.292 *p* = 0.006, 0.034, 0.007, 0.065, 0.065	T4 > T3, T2, Missing > T1

**Table 8 cancers-14-04321-t008:** Patient and tumor characteristics significantly predicting overall survival amongst oral cavity cancer patients by multivariable Cox proportional hazard model (*n* = 282).

Variable/Factor	Statistic	Increased Risk of Death
Age (Continuous)	HR = 0.039 SE = 0.007 *p* < 0.0001	Older age
Sex	HR = −0.514 SE = 0.153 *p* = 0.001	Male > Female
Smoking Status	Base comparison to: Current smoker (vs. ex, never, Missing) HR = −0.634, −0.371, −0.246 SE = 0.196, 0.196, 0.219 *p* = 0.007, 0.001, 0.058, 0.260	Current smoker, Missing > Ex-smoker
Treatment	Base comparison to: all three modalities HR = 0.577, 0.222, −0.638, −0.126 SE = 0.272, 0.292, 0.256, 0.246 *p* < 0.0001, 0.034, 0.447, 0.013, 0.608	Radiotherapy > Surgery/Radiotherapy/Chemotherapy, Surgery

**Table 9 cancers-14-04321-t009:** Patient and tumor characteristics significantly predicting disease-specific (cancer-specific) survival amongst oral cavity cancer patients by multivariable Cox proportional hazard model. The initial model included all those variables significant by univariable.

Variable/Factor	Statistic	Increased Risk of Death
Sex	HR = −0.459 SE = 0.179 *p* = 0.010	Male > Female
Age (Continuous)	HR = 0.028 SE = 0.008 *p* < 0.0001	Older age
Smoking Status	Base comparison to: Current smoker (vs. ex, never, Missing) HR = −0.880, −0.247, −0.086 SE = 0.255, 0.226, 0.243 *p* = 0.005, 0.001, 0.125, 0.725	Current smoker, Never smoker, Missing > Ex-smoker
Treatment Type	Base comparison to: all three modalities HR = 0.840, 0.274, −0.458, −0.051 SE = 0.312, 0.341, 0.303, 0.292 *p* < 0.0001, 0.007, 0.422, 0.131, 0.860	Radiotherapy > Radiotherapy/Chemotherapy, Surgery/Radiotherapy, Surgery/Radiotherapy/Chemotherapy > Surgery

**Table 10 cancers-14-04321-t010:** Patient and tumor characteristics significantly predicting overall survival amongst laryngeal cancer patients by multivariable Cox proportional hazard model (n = 306).

Variable/Factor	Statistic	Increased Risk of Death
Age (Continuous)	HR = 0.030 SE = 0.007 *p* < 0.0001	Older age
TNM Stage	Base comparison to: IV (vs. III, II, I, Missing) HR = −0.704, −1.260, −0.790, −0.653 SE = 0.190, 0.197, 0.189, 0.188 *p* < 0.0001, 0.001, 0.0001, 0.0001, 0.001	IV > II, II, Missing > I
Marital Status	Base comparison to: Single (vs. separated, divorced, married) HR = 0.341, −0.171, 0.340 SE = 0.294, 0.172, 0.200 *p* = 0.008, 0.246, 0.318, 0.090	Single > Married

**Table 11 cancers-14-04321-t011:** Patient and tumor characteristics significantly predicting disease-specific (cancer-specific) survival amongst laryngeal cancer patients by multivariable Cox proportional hazard model. The initial model included all significant variables by univariable.

Variable/Factor	Statistic	Increased Risk of Death
TNM Stage	Base comparison to: IV (vs. III, II, I, Missing) HR = −0.717, −1.423, −0.818, −0.757 SE = 0.264, 0.276, 0.248, 0.248 *p* < 0.0001, 0.007, 0.0001, 0.001, 0.002	Stage IV> Stage III, II, Missing > Stage I

**Table 12 cancers-14-04321-t012:** Patient characteristics indicative of stereotypically HPV-driven oropharyngeal, oral cavity, and laryngeal SCC that may be the basis for the precise selection of patients for whom treatment de-escalation is possible.

Characteristic
HR HPV Positive
Oropharyngeal sub-site
Younger age or ≤50
Never- or ex-smoker

## Data Availability

The data presented in this study are available in this article.

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
