# Peer review of "The Role of HPV in Determining Treatment, Survival, and Prognosis of Head and Neck Squamous Cell Carcinoma"

_cancers, 2022, doi:10.3390/cancers14174321_

Round 1
Reviewer 1 Report
In this manuscript authors performed a retrospective analysis using clinicopathological data of 861 cases of oropharyngeal, oral cavity, and laryngeal SCC. They also analysed FFPE tissue sections for HPV-DNA screening to distinguish between HPV-positive and HPV-negative cases. Data are clearly presented except for tables 4-8 in which authors should uniformize the factors taken into account for the different subsites, they should try to include in each table the same factors for each subsite and in the same order. Moreover, in table 1, I suggest to put the number of HPV-positive cases for each subsite. This is important since authors in line 235 state that there is no difference in the survival between HPV-positive and HPV-negative in the laryngeal and oral cavity SCC cases, but it is not specified how many cases were taken into account, considering that mainly OPSCC are more prone to be HPV-positive.
Thus, also at the beginning of the discussion I would specify that only in OPSCC authors found clear evidences that both overall and cancer-specific survival were significantly improved for HPV positive cases, and not in all the SCC cases.
Finally, how authors explain the fact that in HPV-negative cases there is no differences in overall and cancer-specific survival by treatment type?
Author Response
Dear Reviewer 1,
We thank the reviewers for their time spent thoroughly reviewing our manuscript
We have made the suggested changes including those editorial ones requested (dates for REC approval and referencing style). The reviewers comments are specifically addressed
line by line. These are detailed in the table below and track changed on the paper itself.
Please note also the correction to order of senior authors as advised previously.
Reviewer 1
|
Data are clearly presented except for tables 4-8 in which authors should uniformize the factors taken into account for the different subsites, they should try to include in each table the same factors for each subsite and in the same order. |
See now amended Tables 7-12 and the explanation of all independent variables in Table 1. |
|
Moreover, in table 1, I suggest to put the number of HPV-positive cases for each subsite. This is important since authors in line 235 state that there is no difference in the survival between HPV-positive and HPV-negative in the laryngeal and oral cavity SCC cases, but it is not specified how many cases were taken into account, considering that mainly OPSCC are more prone to be HPV-positive. |
This how now been fully added as Table 6. |
|
Thus, also at the beginning of the discussion I would specify that only in OPSCC authors found clear evidences that both overall and cancer-specific survival were significantly improved for HPV positive cases, and not in all the SCC cases. |
This is defined in Lines 389-90. |
|
Finally, how authors explain the fact that in HPV-negative cases there is no differences in overall and cancer-specific survival by treatment type? |
See lines 463-470. |

Reviewer 2 Report
The authors present a novel study that will increase knowledge. In particular, they offer new information about the epidemiology of HNSCC by HPV status in Ireland and they investigate the impact of HPV status on outcomes in a large cohort with standardized HPV testing. I recommend some improvements in thoroughness of the methods section and in clarity of the information presented in the results tables.
- Lines 23-47: The simple summary identifies the purpose of the study as investigating “HPV’s impact on HNSCC patient outcomes,” (line 27) while the abstract states that the purpose of the study was “to investigate the epidemiology of these HPV-related cancers” (line 36). I would recommend consistency when stating the purpose(s) of the study. You could consider stating epidemiology and outcomes as co-primary purposes.
- Since you mention epidemiology as a purpose of the study, I would recommend including some comparison, likely in the discussion, of the epidemiology in your Irish cohort vs. other cohorts. E.g., does this study suggest that HPV-related disease comprises a greater or lesser proportion of HNSCC cases than in other geographies? You could consider using IARC data (doi 10.1002/ijc.30716).
- Lines 23-47: You could consider being more explicit about your contention that is the largest study of HNSCC outcomes using consistent HPV testing techniques, even beyond Ireland.
- Lines 23-47: It may be useful to note the value of studying historical samples from an era prior to the widespread incorporation of HPV status as a consideration in management decisions. This allows for a prognostic comparison of HPV vs non-HPV disease without differential management directed by HPV status as a confounder.
- Line 28: would change to “appears to be associated with improved survival” (i.e., HPV infection does not improve survival, but HPV-associated disease is associated with longer survival than non-HPV-associated disease)
- Line 43: would change to “surgery alone was associated with prolonged survival”. Although you have conducted multivariable analysis, there is still substantial reason to doubt a causal relationship between surgery alone and survival, given the retrospective nature of this study. For example, I expect that those treated with surgery alone were more likely to have shown negative margins on pathology, and might thus be expected to have better prognosis.
- Lines 46-47: In the introduction, I would suggest stating current HPV vaccination guidelines/practices in Ireland for those unfamiliar. I.e., at what ages are HPV vaccines recommended/provided for girls and boys in Ireland? The text as written makes it seem like boys are not currently being vaccinated for HPV in Ireland — is that correct?
- Lines 180-200: Throughout the manuscript, consider switching the term “multivariate” (denoting multiple dependent variables) to “multivariable” (denoting multiple independent variables). Similarly, consider switching “univariate” to “univariable.”
- Lines 180-200: I would recommend including a comprehensive list (perhaps as a table) of all the independent variables that you tested. It is unclear from the manuscript, as written, whether there were independent variables that were tested but not mentioned in the manuscript since they were not significant on univariable analysis. For example, since “N” stage is not included in any of the tables, it is unclear whether this was an independent variable that you tested (until the reader reaches the discussion, at which point “N” stage is addressed). If you choose to introduce a new table, I would recommend clarifying how each variable was defined (e.g., continuous, ordered/ordinal categorical, unordered categorical) (e.g., T3, T2, and “missing” were grouped into one category).
- Table 3: Strongly recommend reporting here (or somewhere in the body of the text) the number of cases in the sample that were HR HPV positive and the number that were HR HPV negative. Would also report these values for each sub-site.
- Tables 4-9: When HRs, SEs, and P-values are included as a list, it is unclear what each represents. For example, in Table 4, in the second column of the “T stage” row, there are four HR values, four SE values, and five P-values. The comparison to which each of these values corresponds is unclear. From column 3, it appears that three unordered categories are being compared, but this is also not clear.
- Lines 250-266: As stated previously, recommend rephrasing to moderate assertions of causal relationships. For example, the statement “radiotherapy minimized survival” is misleading. Wording like, “Treatment with radiotherapy alone was associated with decreased survival” would be preferable.
- Lines 314-317: Here again, I would be conscious of implying that this study demonstrates a causal relationship between surgery alone and improved survival. Instead of writing “was maximized by surgery alone,” you might consider writing something like “was longest among those treated with surgery alone.”
- Lines 347-367: If I understand correctly, when assigning each patient a TNM stage, you used the 5th edition AJCC system for all patients in this study — is that right? I would recommend including this information in the methods section.
- Line 378: Recommend defining deprivation score/social deprivation in the methods section
- Lines 426-429: Would remove p16 from this sentence. Since p16 status was not included in this study, this study does not indicate anything about whether p16 will play a significant role in determining treatment options.
Author Response
Dear Reviewer 2,
We thank the reviewers for their time spent thoroughly reviewing our manuscript
We have made the suggested changes including those editorial ones requested (dates for REC approval and referencing style). The reviewers comments are specifically addressed
line by line. These are detailed in the table below and track changed on the paper itself.
Please note also the correction to order of senior authors as advised previously.
Reviewer 2
|
Lines 23-47: The simple summary identifies the purpose of the study as investigating “HPV’s impact on HNSCC patient outcomes,” (line 27) while the abstract states that the purpose of the study was “to investigate the epidemiology of these HPV-related cancers” (line 36). I would recommend consistency when stating the purpose(s) of the study. You could consider stating epidemiology and outcomes as co-primary purposes. Since you mention epidemiology as a purpose of the study, I would recommend including some comparison, likely in the discussion, of the epidemiology in your Irish cohort vs. other cohorts. E.g., does this study suggest that HPV-related disease comprises a greater or lesser proportion of HNSCC cases than in other geographies? You could consider using IARC data (doi 10.1002/ijc.30716). |
Lines 36-37. Amended as requested in the tracked changes. |
|
Lines 23-47: You could consider being more explicit about your contention that is the largest study of HNSCC outcomes using consistent HPV testing techniques, even beyond Ireland. |
Lines 36-37. Amended as requested in the tracked changes. |
|
Lines 23-47: It may be useful to note the value of studying historical samples from an era prior to the widespread incorporation of HPV status as a consideration in management decisions. This allows for a prognostic comparison of HPV vs non-HPV disease without differential management directed by HPV status as a confounder. |
Lines 97-99. |
|
Line 28: would change to “appears to be associated with improved survival” (i.e., HPV infection does not improve survival, but HPV-associated disease is associated with longer survival than non-HPV-associated disease) |
Line 28. |
|
Line 43: would change to “surgery alone was associated with prolonged survival”. Although you have conducted multivariable analysis, there is still substantial reason to doubt a causal relationship between surgery alone and survival, given the retrospective nature of this study. For example, I expect that those treated with surgery alone were more likely to have shown negative margins on pathology, and might thus be expected to have better prognosis. |
Line 44. |
|
Lines 46-47: In the introduction, I would suggest stating current HPV vaccination guidelines/practices in Ireland for those unfamiliar. I.e., at what ages are HPV vaccines recommended/provided for girls and boys in Ireland? The text as written makes it seem like boys are not currently being vaccinated for HPV in Ireland — is that correct? |
This has been added into the discussion rather than abstract as it possibly fits better there – see line 558. |
|
Lines 180-200: Throughout the manuscript, consider switching the term “multivariate” (denoting multiple dependent variables) to “multivariable” (denoting multiple independent variables). Similarly, consider switching “univariate” to “univariable.” |
See all mentions to multi and univariate now changed. |
|
Lines 180-200: I would recommend including a comprehensive list (perhaps as a table) of all the independent variables that you tested. It is unclear from the manuscript, as written, whether there were independent variables that were tested but not mentioned in the manuscript since they were not significant on univariable analysis. For example, since “N” stage is not included in any of the tables, it is unclear whether this was an independent variable that you tested (until the reader reaches the discussion, at which point “N” stage is addressed). If you choose to introduce a new table, I would recommend clarifying how each variable was defined (e.g., continuous, ordered/ordinal categorical, unordered categorical) (e.g., T3, T2, and “missing” were grouped into one category). |
See addition of Table 1 Line 185. |
|
Table 3: Strongly recommend reporting here (or somewhere in the body of the text) the number of cases in the sample that were HR HPV positive and the number that were HR HPV negative. Would also report these values for each sub-site. |
See Table 6 Line 255. |
|
Tables 4-9: When HRs, SEs, and P-values are included as a list, it is unclear what each represents. For example, in Table 4, in the second column of the “T stage” row, there are four HR values, four SE values, and five P-values. The comparison to which each of these values corresponds is unclear. From column 3, it appears that three unordered categories are being compared, but this is also not clear. |
See amendments now to Tables 7-12. |
|
Lines 250-266: As stated previously, recommend rephrasing to moderate assertions of causal relationships. For example, the statement “radiotherapy minimized survival” is misleading. Wording like, “Treatment with radiotherapy alone was associated with decreased survival” would be preferable. |
See line 313. |
|
Lines 314-317: Here again, I would be conscious of implying that this study demonstrates a causal relationship between surgery alone and improved survival. Instead of writing “was maximized by surgery alone,” you might consider writing something like “was longest among those treated with surgery alone.” |
See line 405 |
|
Lines 347-367: If I understand correctly, when assigning each patient a TNM stage, you used the 5th edition AJCC system for all patients in this study — is that right? I would recommend including this information in the methods section. |
See Table 1 Line 185 now detailing this. |
|
Line 378: Recommend defining deprivation score/social deprivation in the methods section |
See Table 1 Line 185 now detailing this. |
|
Lines 426-429: Would remove p16 from this sentence. Since p16 status was not included in this study, this study does not indicate anything about whether p16 will play a significant role in determining treatment options. |
Has now been removed from what is now Line 526. |
Imogen

Round 2
Reviewer 1 Report
Authors have answered to all the raised questions.